# Widening Geographical Inequities in DTP Vaccination Coverage and Zero-Dose Prevalence Across Nigeria: An Ecological Trend Analysis (2018–2024)

**DOI:** 10.3390/vaccines13111135

**Published:** 2025-11-04

**Authors:** Hadiza Joy Umar, Solomon Inalegwu Onah, Olalekan Popoola, Hadiza Hussayn Jibril, Femi Oyewole

**Affiliations:** 1Sydani Fellowship Program, Sydani Group, Abuja 900211, Nigeria; hadizajoyumar@gmail.com; 2Center for Vaccine Innovation and Access, PATH, Abuja 900108, Nigeria; hadeezahm@gmail.com; 3Optimising Integrated Campaigns, Acasus, Abuja 900108, Nigeria; lekanpopson16@yahoo.com; 4Independent Consultant, Lagos 105101, Nigeria; femiabedostill@gmail.com

**Keywords:** immunisation equity, zero-dose children, routine immunisation, Nigeria

## Abstract

**Background/Objectives:** Nigeria continues to face major challenges in achieving equitable immunisation coverage, with marked subnational disparities. This study aimed to assess trends in vaccine access and utilisation across Nigeria’s six geopolitical zones between 2018 and 2024, focusing on inequities in DTP coverage, dropout rates, and zero-dose prevalence. **Methods:** We conducted a comparative ecological analysis using secondary data from the Nigeria Demographic and Health Surveys (2018, 2024) and the 2021 Multiple Indicator Cluster Survey/National Immunisation Coverage Survey. Geometric mean coverage for penta 1 (DTP1) and penta 3 (DTP3), DTP1–DTP3 dropout rates, and zero-dose prevalence were calculated for each of the six geopolitical zones and analysed using WHO’s Health Equity Assessment Toolkit Plus. Absolute (difference, D) and relative (ratio, R) summary measures of inequality were also assessed. **Results:** Findings revealed statistically significant differences in indicators across the various regions during the period of study. While the South-East maintained >90% DTP1 coverage, the North-West declined from 37.3% (2018) to 33.4% (2024). In the same period, the absolute inequality (D) in DTP1 coverage increased from 55.3 to 58.4 percentage points. Zero-dose inequities worsened sharply: prevalence in the North-West rose from 25.7% (2021) to 47.4% (2024) compared to ~4% in the South-East, with a relative inequality (R) of 11.29 in 2024. In contrast, service utilisation improved, as dropout rates in the North-West fell from 38.7% (2018) to 14.3% (2024), reducing absolute inequality to 11.0 pp. **Conclusions:** Despite progress in reducing dropout, access to vaccination services remains highly inequitable, particularly in northern Nigeria. Declines since 2021 suggest systemic fragility compounded by COVID-19-related disruptions. Strengthening sustainable routine immunisation systems and investing in demand generation, especially through social and behaviour change communication, are essential to achieving equity.

## 1. Introduction

Vaccination is one of the most efficient and cost-effective public health interventions (second only to ensuring access to clean water) and prevents an estimated 5.1 million deaths from vaccine-preventable diseases each year [1,2,3]. The last two decades have witnessed significant global efforts to enhance vaccination coverage and equity. Initiatives such as the United Nations Sustainable Development Goals (SDGs), Gavi, the Vaccine Alliance, and the Global Vaccine Action Plan 2011–2020 (GVAP) have driven substantial progress in increasing access to and utilisation of vaccination services. However, global coverage for the third dose of diphtheria–tetanus–pertussis vaccine (DTP3), a key indicator of immunisation system performance, has stagnated at approximately 85% since 2010, falling short of the 90% target achieved by only 64% of countries. This stagnation underscores persistent and significant variability in immunisation coverage, both between and within nations, leaving specific populations disproportionately under-immunised [1].

While the Expanded Programme on Immunisation (EPI) has achieved remarkable progress in improving access to routine childhood vaccines across Africa, Nigeria presents a particularly stark and persistent challenge. Using survey data, reports of DTP3 coverage in Nigeria were critically low at 15% in 1995 and reached 33% by 1999 before declining to 28.1% in 2007 [4,5,6]. National and global efforts to strengthen Nigeria’s EPI programme led to an increase in DPT3 coverage to 43% between 2006 and 2011. However, coverage sharply dropped to 34.4% by 2016 due to systemic challenges and insecurity [7,8]. While recent improvements pushed coverage to 56.6% in 2021 and 53.5% in 2023/24 [9,10], these rates remain below the 80% target and reflect persistent barriers like access issues and vaccine hesitancy. This profound inequity in access to vaccination services means that Nigeria alone accounts for 14.7% of the world’s 14.3 million unvaccinated infants in 2024 [11], creating a vast pool of susceptible children that contributes to the high burden of under-5 mortality, thereby fuelling the morbidity and mortality from vaccine-preventable disease outbreaks within the country.

Vaccination inequity refers to the lack of equitable distribution or the unfair distribution, access to, and utilisation of vaccination services across population groups [12]. Notably, there is considerable variance in vaccination access and utilisation between and within regions and states in Nigeria, usually correlating with the economic, cultural, and behavioural factors of the population groups affected [13]. For instance, Jean Batiste et al. reported that the prevalence of zero-dose and under-immunised children was primarily driven by social inequity, with higher rates observed in northern Nigeria [14]. In their 2024 review of reports of enablers and barriers to vaccine uptake in Nigeria, Mohammed and colleagues found that the low uptake in North-Western and North-Central states was primarily culturally driven by husband-headed households. Consequently, husbands usually made decisions about vaccination for their children. They noted that the decision to vaccinate was either explicitly or indirectly expressed by refusal to cover indirect costs and provide necessary support to the mothers [15]. Their study also found that in the northern region of the country, the engagement and mobilisation of community influencers (such as religious and traditional leaders) as vaccine advocates significantly improved uptake of vaccination services. Conversely, they found that in the southern region of the country, maternal autonomy in decision-making, health literacy, widespread use of reminder systems (e.g., SMS and WhatsApp), and the belief that vaccination defines responsible parenting were key factors that drove higher uptake of vaccination services, a finding echoed by Alabi et al. [16] and Olufadewa et al. [17].

Ogundele et al. reported that the goal of the National Immunisation Programme in Nigeria, which is to eradicate childhood diseases and improve the health of Nigerian children, is threatened by suboptimal uptake of vaccines despite the established benefits of vaccines [18]. They also pointed out that polio vaccine refusal in Northern Nigeria in 2003/2004 not only quintupled the polio incidence in Nigeria, but also contributed to outbreaks across three continents. 

This study attempts to quantify and identify trends in vaccination equity in Nigeria with the aim of generating actionable recommendations to bridge this gap.

## 2. Materials and Methods

### 2.1. Study Design and Setting

This review presents an ecological trends analysis of the diphtheria-, tetanus-, and pertussis-containing vaccine (DTP) coverage rates and vaccination inequity across the states in Nigeria for the period 2018 to 2024. The states are categorised into northern and southern regions, and further segmented into 6 geopolitical zones based on General Sani Abacha’s administrative grouping of the country’s states [19]. In this study, the Federal Capital Territory is treated as the 37th state in the country. Figure 1 visualises the 6 geopolitical zones.

### 2.2. Data Collection and Data Analysis

Secondary data was gathered and analysed to evaluate equity in vaccination access and utilisation across Nigeria, utilising data from 2018 to 2024. State-level data on vaccination coverage in Nigeria was extracted from the 2018 Nigeria Demographic and Health Survey (NDHS) [20], 2021 Multiple Indicator Cluster Survey & National Immunisation Coverage Survey (MICS/NICS) [9], and the 2023–24 NDHS [10]. The similarity in survey architecture and weighting strategies used in these surveys provides a basis for comparing vaccination coverage trends over time. Appendix A presents the vaccination coverage data compiled from all three sources.

Various data analysis tests (defined below) were conducted, including the calculation of vaccination dropout rates and summary measures of inequality such as differences and ratios. The analysis employed Microsoft Excel 2019, IBM SPSS Statistics 27, and the WHO Health Equity Assessment Tool Plus (Upload Edition). Appendix A contains the dataset utilized for the analyses performed using the WHO Health Equity Assessment Tool.

### 2.3. Definition of Selected Indicators

Vaccination coverage: Vaccination coverage refers to the number of persons belonging to a target population who are vaccinated against a specific disease, divided by the total number of individuals belonging to the same population [21]. This review primarily focuses on DTP-containing vaccination coverage, a widely used indicator, to assess immunisation programme performance and healthcare system vulnerabilities [22,23]. Since 2012, the DTP-containing vaccine used in Nigeria has been the pentavalent vaccine (commonly referred to as penta), replacing the standalone DTP and HepB vaccines [24].Vaccination dropout rate: Vaccination dropout rate refers to the proportion of vaccine recipients who do not complete their vaccination schedules for a multidose vaccine and it is determined by calculating the percentage of vaccinees who completed the schedule compared to those who started it [25]. In the absence of reliable subnational-level longitudinal vaccination data, dropout rates calculated from cross-sectional surveys have been used to as a stand-in and have been used to infer programme performance. In this review, it is calculated as the proportion of recipients who received the DTP1 vaccine (access to immunisation system) against the proportion that received the DTP3 vaccine (utilisation of the system).Zero-dose children: Zero-dose children refer to children who have not received any routine vaccine on the Expanded Programme on Immunisation schedule [26]. The proportion of children who have not received the first dose of the DTP-containing vaccine (DPT1) is commonly used as a proxy for the proportion of zero-dose children in a population [27].Ratio (*R*): The ratio is a relative measure of health inequality, calculated as the quotient of indicator values between two population subgroups [28]. It is expressed using the formula below.


$$R=\frac{y_{1}}{y_{2}}$$


It should be noted that the selection of *y*_1_ and *y*_2_ depends on the characteristics of the inequality dimension and the type of indicator for which *R* is calculated. An overview of the calculation is detailed in the technical notes of the Health Equity Assessment Toolkit. If there is no inequality, *R* takes the value one. The further the value of *R* from one, the higher the level of inequality [28].

Difference (*D*): Difference is an absolute measure of inequality that shows the range of indicator values between two population subgroups [28]. It is expressed using the formula below.


$$D=y_{1}-y_{2}$$


The selection of *y_1_* and *y_2_* also depends on the characteristics of the inequality dimension and the type of indicator, for which *D* is calculated. An overview of the calculation is detailed in the technical notes of the Health Equity Assessment Toolkit [28]. If there is no inequality, *D* takes the value zero. Greater absolute values indicate higher levels of inequality.

Mean difference from best performing subgroup (unweighted) (MDBU): MDBU is an absolute and complex measure of health inequality that indicates the unweighted mean difference between each population subgroup and the best-performing subgroup [28]. It is calculated using the formula below.


$$MDBU=\frac{1}{n}\times \sum_{i}\left|y_{i}-y_{best}\right|$$


An overview of the calculation is detailed in the technical notes of the Health Equity Assessment Toolkit [28]. MDBU is zero if there is no inequality, with larger values indicating higher levels of inequality.

Vaccine access: Vaccine access refers to the system-level capacity to provide vaccines within reach of populations. It is usually measured by the vaccination coverage for the DTP1 vaccine [29]. A high DTP1 coverage indicates high access to routine immunisation services for the examined population. In Nigeria, the country aims to achieve 80% coverage for routine immunisations across the country [30].Vaccine utilisation: Vaccine utilisation measures the uptake of vaccine services and is measured by vaccination dropout rates between the early and final doses [30]. A dropout rate of >10% is unfavourable and indicates that a health facility has limitations in utilisation [31].Kruskal–Wallis H Test: The Kruskal–Wallis H test is a nonparametric test that can be used to determine if there are statistically significant differences between two or more groups of an independent variable (i.e., geopolitical regions) on a continuous dependent variable (i.e., vaccination coverage, vaccination dropout, zero-dose prevalence) [32].

## 3. Results

### 3.1. Trends in Subnational Penta 1 Vaccination Equity in Nigeria (2018–2024)

Performing a Kruskal–Wallis test revealed statistically significant differences in penta 1 vaccination coverage across the country’s six geopolitical zones in 2018, 2021, and 2024 (*p* < 0.001 for all years; 2018 Kruskal–Wallis H = 26.40, 2021 Kruskal–Wallis H = 24.21, 2024 Kruskal–Wallis H = 25.13). Further analysis of the geometric means of penta 1 coverage across the country’s six geopolitical regions demonstrates this persistent and widening equity gap between 2018 and 2024, as shown in Table 1. A stark north–south divide characterised the observed periods, with southern regions maintaining consistently higher vaccination rates compared to their northern counterparts. The South-East remained the highest-performing region throughout the study period, recording 92.6% coverage in 2018, 92.3% in 2021, and 91.8% in 2024. Similarly, the South-South showed steady improvement from 84.6% to 87.8% over the same period. The South-West exhibited moderate volatility, declining from 86.4% in 2018 to 81.6% in 2021 before partially recovering to 83.8% in 2024.

In contrast, northern regions displayed concerning trends, particularly after 2021. As visualised in Figure A1, Figure A2 and Figure A3, while the North-East demonstrated modest but consistent gains, rising from 54.7% in 2018 to 64.4% in 2024, the North-Central and North-West experienced severe reversals. The North-Central declined sharply from 76.2% in 2021 to 60.5% in 2024, falling below its 2018 baseline of 73.9%. Most alarmingly, the North-West—already the lowest-performing region in 2018 (37.3%)—regressed to 33.4% in 2024 after a temporary improvement to 51.8% in 2021, representing an 18.4-percentage-point decline since 2021.

### 3.2. Trends in Subnational Penta 3 Vaccination Equity in Nigeria (2018–2024)

The Kruskal–Wallis test confirmed statistically significant differences in penta 3 vaccination coverage across the country’s six geopolitical zones in 2018, 2021, and 2024 (*p* < 0.001 for all years; 2018 Kruskal–Wallis H = 25.37, 2021 Kruskal–Wallis H = 22.87, 2024 Kruskal–Wallis H = 22.14). Additionally, the characteristic north–south disparity observed for penta 1 persisted for penta 3, though with distinct regional trajectories and widening absolute gaps in recent years. As shown in Table 2, the southern regions maintained substantially higher coverage throughout the period, with the South-South achieving the highest 2024 coverage at 79.9%, followed closely by the South-East at 79.8%. The South-West showed moderate volatility, declining from 70.3% in 2018 to 66.5% in 2021 before partially recovering to 69.3% in 2024.

Northern regions exhibited divergent patterns, with the North-East demonstrating the most substantial improvement, rising from 36.5% in 2018 to 52.6% in 2024, a 16.1-percentage-point gain. In contrast, the North-Central declined steadily from 57.9% in 2018 to 47.7% in 2024, representing a 10.2-percentage-point decrease, as seen in Figure A4, Figure A5 and Figure A6. Most critically, the North-West remained the lowest-performing region, declining sharply to 28.0% in 2024 after a temporary gain to 36.2% in 2021.

### 3.3. Trends in Subnational Penta 1–3 Dropout Rate Equity in Nigeria (2018–2024)

The Kruskal–Wallis test revealed statistically significant differences in penta 1–3 dropout rates across the Nigeria’s six geopolitical zones for in 2018 (*p* = 0.005, Kruskal–Wallis H = 16.77) and 2021 (*p* = 0.008, Kruskal–Wallis H = 15.52). As indicated in Table 3, analysis of geometric mean penta 1–3 dropout rates across Nigeria’s geopolitical regions reveals important improvements in the continuity of vaccination services, though with persistent regional disparities. The data demonstrates a notable reduction in dropout rates across most regions between 2018 and 2024, particularly in historically disadvantaged northern areas, while revealing emerging challenges in other regions.

Northern regions showed substantial progress over the study period. The North-West achieved the most dramatic improvement (visualised in Figure A7, Figure A8 and Figure A9), reducing its dropout rate from 38.7% in 2018 to 14.3% in 2024, a 24.4-percentage-point decline. Similarly, the North-East decreased from 31.7% to 15.3%, representing a 16.4-percentage-point improvement. The North-Central region demonstrated more modest gains, moving from 20.0% in 2018 to 19.0% in 2024, though this region experienced a temporary increase to 21.3% in 2021.

Southern regions displayed mixed trends. The South-South achieved the lowest dropout rate nationally in 2024 (8.0%), improving substantially from 18.5% in 2018. In contrast, the South-East experienced a concerning increase from 9.9% in 2018 to 11.6% in 2024, despite reaching a low of 5.0% in 2021. The South-West similarly deteriorated, rising from 10.3% in 2018 to 15.0% in 2024, representing the second-highest regional dropout rate in the latest survey.

### 3.4. Trends in Subnational Zero-Dose Children Inequity in Nigeria (2018–2024)

The Kruskal–Wallis test confirmed statistically significant differences in subnational zero-dose prevalence across the country’s six geopolitical zones in 2018, 2021, and 2024 (*p* < 0.001 for all years; 2018 Kruskal–Wallis H = 22.29, 2021 Kruskal–Wallis H = 17.96, 2024 Kruskal–Wallis H = 23.91). Further analysis of geometric mean zero-dose child prevalence across Nigeria’s geopolitical regions reveals alarming reversals of progress and intensifying geographical inequities between 2018 and 2024. Table 4 reveals that the northern regions have experienced catastrophic increases in unvaccinated children since 2021, completely erasing earlier gains and expanding the north–south divide to unprecedented levels. This trend is visualised in Figure A10, Figure A11 and Figure A12.

The North-West emerged as the most critically affected region, with zero-dose prevalence surging from 25.7% in 2021 to 47.4% in 2024—nearly doubling in just three years and exceeding its 2018 baseline of 28.4%. Similarly, the North-Central experienced a devastating 17.0-percentage-point increase from 2021 to 2024 (9.6% to 26.6%), more than tripling its rate since 2018 (12.2%). While the North-East showed a more moderate increase (21.7% to 25.0%), it still represented substantial regression from 2018 (23.4%).

Southern regions maintained substantially lower zero-dose prevalence, though with concerning trends in the South-West. The South-East remained the strongest performer (4.5% in 2018 to 4.2% in 2024), while the South-South improved from 9.1% to 6.2%. However, the South-West more than doubled its prevalence from 7.2% in 2021 to 10.2% in 2024, positioning it as the highest-burden southern region.

### 3.5. Summary Measures of Inequality

Analysis of absolute and relative inequality measures for key immunisation indicators reveals divergent equity trajectories across Nigeria. The difference (D), ratio (R) and mean difference from the best performing subgroup (unweighted) (MDBU) metrics demonstrate major fluctuations in geographical disparities over the study period, with concerning reversals in progress in recent years.

For penta 1 coverage, absolute inequality (D) between the best and worst performing regions decreased from 55.3 percentage points in 2018 to 40.5 in 2021, suggesting narrowing geographical gaps. However, this trend reversed dramatically by 2024, when D surged to 58.4 percentage points, exceeding the 2018 baseline and representing the highest recorded disparity. Taking all the regions into consideration, the MDBU also showed the same pattern of initial improvement from 21.0 percentage points in 2018 to 17.4 in 2021 before a subsequent decline to 21.5 percentage points in 2024. Relative inequality (R) followed a similar pattern, improving from 2.48 in 2018 to 1.78 in 2021 before deteriorating to 2.75 in 2024. This indicates that in 2024, the highest-performing region had nearly triple the coverage of the lowest-performing region for the first dose of the pentavalent vaccine.

For penta 3 coverage, moderate improvements occurred between 2018 and 2021, with absolute inequality (D) declining from 61.7 to 48.6 percentage points, and relative inequality (R) decreasing from 4.02 to 2.34. However, these gains were partially reversed by 2024, as D increased to 51.9 and R rose to 2.85. The 2024 R value indicates that the highest-performing region had almost three times greater coverage than the lowest-performing region for the third dose of the pentavalent vaccine. When all regions were taken into account, however, the MDBU showed a consistent reduction in inequality in penta 3 vaccine coverage across regions, improving from 26.3 percentage points in 2018 to 20.4 in 2024.

For the proportion of zero-dose children, both absolute and relative inequality measures show accelerating geographical disparities. Absolute inequality (D) increased from 23.9 percentage points in 2018 to 43.2 in 2024, with the most dramatic surge occurring between 2021 and 2024 (22.8 to 43.2). Relative inequality (R) exhibited even steeper deterioration, rising from 6.31 in 2018 to 11.29 in 2024. The 2024 R value indicates that the worst-performing region had over 11 times more zero-dose prevalence than the best-performing region, the largest relative disparity among all indicators. The MDBU followed a similar trend for this indicator, showing a slight improvement from 9.5 percentage points in 2018 to 9.4 in 2021, before worsening to 15.7 percentage points in 2024.

The penta 1–3 dropout rate consistently improved throughout the study period. Absolute inequality (D) declined substantially from 28.8 percentage points in 2018 to 11.0 in 2024. Relative inequality (R) showed parallel improvement, decreasing from 3.91 to 2.38. The 2021 temporary increase in R (5.34) was subsequently overcome, reflecting sustained progress in reducing geographical disparities for this indicator. The MDBU confirmed this reduction in inequality across the country for the penta 1–3 dropout rate from 11.6 percentage points in 2018 to 5.9 percentage points in 2024. These changes are tabulated in Table 5.

## 4. Discussion

The trend analysis demonstrates that the challenge faced by the EPI programme in Nigeria is more of an issue of access to vaccination services than it is of utilisation of vaccination services. Save for the North-Central region, the results show an overall improvement and/or stabilisation in the vaccination utilisation indicator and a reduction in dropout rates across the country. However, the equity gap for access to vaccination services between extreme-performing regions (measured using penta 1 coverage) widened substantially over time. The increasing North-West–South-East differential in penta 1 vaccine coverage between 2018 and 2024, despite improvements reported in 2021, reflects a worrying instance of accelerating inequity that offsets successes made across various indicators.

The divergence also represents a concerning reversal of some of the initial progress made following the launch of the National Emergency Routine Immunisation Coordination Committee (NERICC), indicating stagnation or even localised deterioration in access improvements post-2021, particularly in the North-West and North-Central regions of the country. As reported by the WHO, NERICC was established in 2017, prompted by a national RI coverage of 33% in 2016 (based on the 2016/2017 MICS/NICS report) [33]. The initiative prioritised 18 low-performing states for implementation of strategic interventions (Sokoto, Jigawa, Kaduna, Kano, Katsina, Borno, Gombe, Bauchi, Adamawa, Yobe, Zamfara, Kebbi, Kogi, Taraba, Nasarawa, Niger, Bayelsa, and Plateau states), most of which were in the northern region of the country. The WHO also reported that this focus yielded major gains, increasing the national-level RI coverage to 54% in 2018 (according to the 2018 SMART survey) and 56.6% by 2021, with substantial improvements across all regions of the country. The establishment of NERICC also brought about a slight reduction in the zero-dose burden, especially in the North-Central and North-Western regions of the country, indicating that access to vaccination services was indeed improved during this period.

A potential catalyst for the post-2021 dip in vaccination access indicators in the North-West and North-Central regions of the country is likely the COVID-19 pandemic-driven interruptions in supplementary immunisation activities (SIAs), which have been the central measure for bolstering vaccination programmes in the NERICC states. Omoleke and their team [34], analysing the trend in RI uptake in Kebbi state (a NERICC state) between January and September 2019, found that the integration of RI vaccination into supplementary immunisation activities (SIAs) improved RI coverage and reduced dropout rates, with up to 70% of children who received the RI vaccines receiving it during the SIA campaign. However, the same study found that the trend in the uptake and utilisation of these vaccination services was strongly correlated with the implementation of the SIA campaigns, which presents the issue of sustainability. This is because the equity gaps in access to and utilisation of vaccination services, which have been reduced during the implementation of these campaign activities, may be short-lived due to the RI service delivery systems being incapable of sustaining their recorded progress.

Haenssgen, Closser, and Alonge [35], in their study of the impact of mass vaccination campaigns in Nigeria, reported that high-frequency implementation of SIAs had detrimental health system effects that potentially left 3.6 million children deprived of full immunisation. They also reported that the frequency of these campaigns was higher in regions with weak health systems, causing further disruptions to the operations of RI service delivery. Their report aligns closely with Cutts et al. [36], who pointed out that multiple countries with weak RI systems favour the SIA approach to compensate for the state of their RI system. Barron et al. [37] also posited that reliance on vaccination campaigns can lead to duplication of activities and inefficient use of resources, as the resources that should have been directed towards strengthening RI service delivery are diverted towards implementing resource-intensive SIA campaigns.

While the NERICC initiative generated measurable gains in immunisation coverage as seen in various national and subnational-level indices, its approach may have ultimately struggled to reinforce the resilience of Nigeria’s RI systems, especially in regions already experiencing vaccination inequity. Future interventions initiated to bridge these equity gaps must prioritise sustainable and resilient demand generation through rigorously tailored social and behaviour change communication (SBCC). This would involve strengthening key aspects of ongoing advocacy, communication, and social mobilisation activities, including the following:Context-specific messaging that addresses region-specific barriers (e.g., vaccine hesitancy in conflict-affected areas, gender-related constraints).Community-led engagement through partnerships with local leaders, religious institutions, and community health workers to foster trust and community ownership.Further integration of the RI programme into other areas of primary health systems, such as embedding SBCC for RI services within primary healthcare revitalisation efforts, ensuring consistent engagement with the target audience.

Without such systemic reorientation, Nigeria’s immunisation equity gaps will continue, rendering these underserved communities vulnerable to shocks and reliant on resource-intensive stopgap campaigns.

## 5. Conclusions

This study exposes critical systemic fragility in Nigeria’s immunisation programme, particularly in the North-West and North-Central regions, where substantial declines in DTP1 coverage and near-doubling of zero-dose prevalence have reversed prior gains. These trends have driven a widening equity gap. To close these gaps, Nigeria must increase investment in contextualised and region-specific SBCC interventions to address the drivers of these inequities. Without these, underserved regions will remain vulnerable and dependent on unsustainable, campaign-based interventions, and Nigeria will continue to risk falling short of its universal health coverage commitments and the equity goals of the Immunisation Agenda 2030. The authors also recommend further investigation of the drivers of these inequities through more granular studies involving the key stakeholders involved in the vaccination programme in Nigeria.

## Figures and Tables

**Figure 1 vaccines-13-01135-f001:**
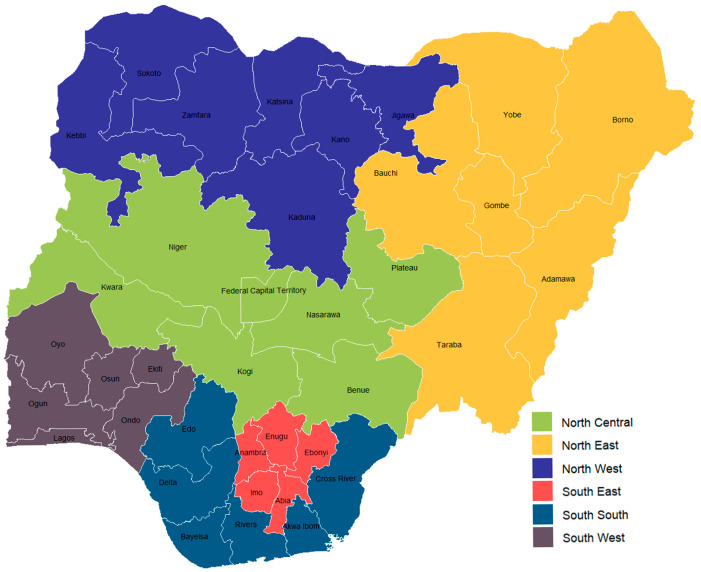
Distribution of Nigerian states across the 6 geopolitical zones.

**Table 1 vaccines-13-01135-t001:** Geometric mean of penta 1 vaccine coverage in Nigeria by region, 2018–2024 (%).

Region	2018	2021	2024	Change (2018–2024)
North-Central	73.9	76.2	60.5	−13.4
North-East	54.7	61.2	64.4	+9.7
North-West	37.3	51.8	33.4	−3.9
South-East	92.6	92.3	91.8	−0.8
South-South	84.6	86.4	87.8	+3.2
South-West	86.4	81.6	83.8	−2.6

**Table 2 vaccines-13-01135-t002:** Geometric mean of penta 3 vaccine coverage in Nigeria by region, 2018–2024 (%).

Region	2018	2021	2024	Change (2018–2024)
North-Central	57.9	58.8	47.7	−10.2
North-East	36.5	43.2	52.6	+16.1
North-West	20.4	36.2	28.0	+7.6
South-East	82.1	84.8	79.8	−2.3
South-South	67.4	74.7	79.9	+12.5
South-West	70.3	66.5	69.3	−1.0

**Table 3 vaccines-13-01135-t003:** Geometric mean of penta 1–3 dropout rate in Nigeria by region, 2018–2024 (%).

Region	2018	2021	2024	Change (2018–2024)
North-Central	20.0	21.3	19.0	−1.0
North-East	31.7	26.7	15.3	−16.4
North-West	38.7	23.1	14.3	−24.4
South-East	9.9	5.0	11.6	+1.7
South-South	18.5	13.1	8.0	−10.5
South-West	10.3	15.3	15.0	+4.7

**Table 4 vaccines-13-01135-t004:** Geometric mean of zero-dose child prevalence in Nigeria by region, 2018–2024 (%).

Region	2018	2021	2024	Change (2018–2024)
North-Central	12.2	9.6	26.6	+14.4
North-East	23.4	21.7	25.0	+1.6
North-West	28.4	25.7	47.4	+19.0
South-East	4.5	2.9	4.2	−0.3
South-South	9.1	6.5	6.2	−2.9
South-West	6.4	7.2	10.2	+3.8

**Table 5 vaccines-13-01135-t005:** Summary measures of inequality in Nigeria by indicator, 2018–2024.

Indicator	Measure	2018	2021	2024
Penta 1 Coverage	Difference	55.3	40.5	58.4
	Ratio	2.48	1.78	2.75
	MDBU	21.0	17.4	21.5
Penta 3 Coverage	Difference	61.7	48.6	51.9
	Ratio	4.02	2.34	2.85
	MDBU	26.3	24.1	20.4
Zero-Dose Children	Difference	23.9	22.8	43.2
	Ratio	6.31	8.86	11.29
	MDBU	9.5	9.4	15.7
Penta 1–3 Dropout Rate	Difference	28.8	21.7	11.0
	Ratio	3.91	5.34	2.38
	MDBU	11.6	12.4	5.9

MDBU: Mean difference from best performing subgroup (unweighted).

## Data Availability

The authors confirm that the data supporting the findings of this study is available within the article and its Appendix A.

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
