# Peer review of "Widening Geographical Inequities in DTP Vaccination Coverage and Zero-Dose Prevalence Across Nigeria: An Ecological Trend Analysis (2018–2024)"

_vaccines, 2025, doi:10.3390/vaccines13111135_

Round 1
Reviewer 1 Report
Comments and Suggestions for Authors
This is an ecological study, written by authors that are not affiliated by a recognized institution, making me wonder about the overall validity. The list of problems include the lack of information on how and where the data were obtained? Why did you use the exact geographical definitions, are these commonly used? Could this have been a source of misalignment? Please provide the exact source of data, with references. The main problem is the lack of proper statistical analysis. Comparisons of two or three percentages is insufficient, since that can not be considered as reliable. You need to expand the analysis somehow, by analysing trends, Poisson, or even linear trend, with significance testing. Only then can you claim that the change was significant or not. I guess, you could even use a chi-squared test to see if the pattern is outside the null-trend. You could also use GIS and do some more refined analyses. Statistics needs improvement anyhow. Delete the first section of Results, this is just an element of the template. Expand by putting this in context, which is even more relevant than the numerical analysis. What does this mean, and what can be done? Also, you need to deal with the fact that this is an ecological design, and suggest smaller-size studies that could fill in the gaps in knowledge and provide evidence where it is direly lacking.
Author Response
Dear Reviewer, thank you very much for your comments and for helping in improving the quality of our work. We have listed a point-by-point response below
- Comment: This is an ecological study, written by authors that are not affiliated by a recognized institution, making me wonder about the overall validity.
Response: The authors of this study are affiliated to recognized institutions that directly work with the government of Nigeria in providing technical assistance and direct implementation support for its routine immunization program. This includes Sydani Group, PATH, and Acasus.
- Comment: The list of problems include the lack of information on how and where the data were obtained?:
Response: Information on how and where the data for this study was obtained is explicitly stated in the "Data Collection and Data Analysis" sub-section of the Method section. The data for this study (State-level data on vaccination coverage in Nigeria) was sourced from the publicly available reports of the 2018 Nigeria Demographic and Health Survey (NDHS), 2021 Multiple Indicator Cluster Survey & National Immunisation Coverage Survey (MICS/NICS), and the 2023–24 NDHS.
- Comment: Why did you use the exact geographical definitions, are these commonly used? Could this have been a source of misalignment?
Response: The exact geographical definition used in the study and the rationale for its use have been explained in the manuscript. The 6 geopolitical zones of Nigeria have been used to categorize the states in the country for administrative purposes for approximately 30 years now are very commonly used. Furthermore, the north-south categorization have been used since colonial times, preceding the formation of Nigeria as a country following the amalgamation of the northern and southern protectorates in 1914. In addition, data from the data sources (MICS/NICS and NDHS) are also categorized in this manner.
- Comment: Please provide the exact source of data, with references.
Response: The exact source of the data and references are already provided in the manuscript in the “Data Collection and Data Analysis” subsection of the methods section.
- Comment: The main problem is the lack of proper statistical analysis. Comparisons of two or three percentages is insufficient, since that can not be considered as reliable. You need to expand the analysis somehow, by analysing trends, Poisson, or even linear trend, with significance testing. Only then can you claim that the change was significant or not. I guess, you could even use a chi-squared test to see if the pattern is outside the null-trend. You could also use GIS and do some more refined analyses. Statistics needs improvement anyhow.
Response: The team has included Kruskal-Wallis tests to improve the reliability of the result and improve the statistics of the work.
- Comment: Delete the first section of Results, this is just an element of the template. Expand by putting this in context, which is even more relevant than the numerical analysis. What does this mean, and what can be done?
Response: The artifact from the template has been removed from the manuscript. The results was also contextualized and discussed extensively in the discussion section.
- Comment: Also, you need to deal with the fact that this is an ecological design, and suggest smaller-size studies that could fill in the gaps in knowledge and provide evidence where it is direly lacking.
Response: A statement recommending more granular studies to address this gap has been included to the conclusion section of the revised manuscript.
We hope these response adequately address your comments so far.
Thank you very much again.
Reviewer 2 Report
Comments and Suggestions for Authors
Dear Authors,
While the topic is timely and highly relevant for global immunization equity, the manuscript presents several substantial limitations.
First, estimates from different surveys (NDHS, MICS/NICS) are combined without adequate methodological harmonization or discussion of sampling, recall bias, or weighting differences, making trend comparisons unreliable.
The analysis reports only point estimates, without confidence intervals, design-based standard errors, or tests of statistical significance. This prevents the reader from assessing whether observed differences are meaningful.
The use of geometric means across states, without population weighting or sensitivity checks, introduces bias. The interpretation of dropout rates from repeated cross-sectional data is also problematic.
Causal links to COVID-19 disruptions or reliance on supplemental immunization activities are suggested without supporting data or triangulation.
I noted that several structural issues (duplicate table numbering, placeholders from the journal template, limited figures) hinder clarity and readability.
In conclusion, I think that in its current form, the manuscript offers descriptive observations but does not provide sufficiently robust or reproducible evidence to advance the scientific debate on vaccination inequities.
Several minor grammar issues and template artefacts.
Author Response
Dear reviewer, thank you very much for your comments and for contributing in improving the quality of our work.
We have outlined a point-by-point response to the comments raised below.
- Comment: First, estimates from different surveys (NDHS, MICS/NICS) are combined without adequate methodological harmonization or discussion of sampling, recall bias, or weighting differences, making trend comparisons unreliable.
Response: We appreciate the reviewer’s concern regarding the comparability of estimates across different surveys (NDHS, MICS/NICS). However, we argue that the methodologies and sampling strategies of these surveys are sufficiently aligned to permit reliable trend analyses. All three surveys (NDHS 2018, MICS/NICS 2021, and NDHS 2023–24) employed stratified, two-stage cluster sampling designs based on enumeration areas derived from the national census cartographic frames. In each case, clusters were selected with probability proportional to size within strata defined by state and urban/rural residence, followed by systematic random sampling of households within clusters. Household listings were freshly conducted before second-stage selection, ensuring updated frames and reducing coverage error. Sample sizes were also broadly comparable across surveys (NDHS ≈ 42,000 households; MICS/NICS ≈ 37,000 + 6,740 supplemental households), and each survey incorporated sampling weights to account for the non-proportional allocation of samples and differential nonresponse, thereby producing estimates representative at the national, zonal, and state levels.
The similarity in survey architecture and weighting strategies provides a solid basis for comparing vaccination coverage trends over time. A statement addressing this concern has also been included in the revised manuscript.
- Comment: The analysis reports only point estimates, without confidence intervals, design-based standard errors, or tests of statistical significance. This prevents the reader from assessing whether observed differences are meaningful.
Response: Tests for statistical significance (Kruskal-Wallis Tests) have been included in the study to complement the reported point estimates and help the readers assess meanings from observed differences.
- Comment: The use of geometric means across states, without population weighting or sensitivity checks, introduces bias.
Response: Geometric means were chosen to calculate averages across the various states in each region, particularly for its less sensitivity to extreme values. The Kruskal-Wallis test for statistical significance, which uses the individual state-level indicator values (instead of the calculated geometric means), buttresses and checks the inferences made from analysis involving the geometric means, hopefully addressing this concern.
- Comment: The interpretation of dropout rates from repeated cross-sectional data is also problematic.
Response: In the absence of reliable subnational-level longitudinal vaccination data, dropout rates calculated from cross-sectional surveys have been used in-country as a stand-in and has been used to infer program performance as is intended in this study. A statement addressing this concern has been included in the revised manuscript.
- Comment: Causal links to COVID-19 disruptions or reliance on supplemental immunization activities are suggested without supporting data or triangulation.
Response: The relationships suggested in this study are consistent with the reports from existing literature as well as the field and programmatic observations in Nigeria’s EPI program implementation. However, included in the conclusion of this study is a recommendation for more granular studies that will more empirically confirm the existence of this relationship.
- Comment: I noted that several structural issues (duplicate table numbering, placeholders from the journal template, limited figures) hinder clarity and readability.
Response: This has been addressed in the revised manuscript. The manuscript has been proofread by an editorial service provider, and figures have been included to improve clarity and readability.
- Comment: In conclusion, I think that in its current form, the manuscript offers descriptive observations but does not provide sufficiently robust or reproducible evidence to advance the scientific debate on vaccination inequities.
Response: The authors have attempted to address this concern by including additional analyses, providing more reproducible evidence to advance the scientific debate on vaccination inequities.
We hope that these responses have addressed your concerns about our work.
Thank you again.
Reviewer 3 Report
Comments and Suggestions for Authors
The manuscript, “Widening Geographical Inequities in DTP Vaccination Coverage and Zero-Dose Prevalence Across Nigeria: An Ecological Trend Analysis (2018–2024)” by Umar et al., evaluates DTP vaccination coverage in various geographical regions of Nigeria between 2018 and 2024. The authors used data sets from Nigeria’s health surveys (NDHS, MICS/NICS) and analyzed the DTP1 and DTP3 coverage in six different geopolitical regions of Nigeria. The study concludes TDP vaccination coverage across various regions of Nigeria remains highly discriminatory which needs to be addressed for better implementation of routine immunization. The manuscript is well written with clearly defined methods and results. Although, authors are suggested to include a map based comparison for more visual clarity of their analysis.
Author Response
We sincerely thank the reviewer for the positive and encouraging feedback on our manuscript. We particularly appreciate the suggestion to enhance the visual clarity of the study through the inclusion of map-based comparisons.
In response, we have incorporated maps (mostly included in the appendix section so as not to interrupt the flow of the document) to strengthen the presentation of regional disparities and improve the overall readability of the results. We believe these additions make the study more intuitive and accessible to a wider audience.
We are grateful for the reviewer’s insightful comments, which have helped us to improve the manuscript.
Reviewer 4 Report
Comments and Suggestions for Authors
A very interesting manuscript entitled ‘Widening Geographical Inequities in DTP Vaccination Coverage and Zero-Dose Prevalence Across Nigeria: An Ecological Trend Analysis (2018–2024)’ was submitted to be considered for its eventual publication in the journal Vaccines. Thus, after having read carefully the manuscript, I consider that it is suitable for its publication practically in its present form due to many positive aspects. For instance, the introductory part shows suitably the context of the work. References cited are enough and pertinent. Materials and methods’ section is described in detail. Results are well discussed. Conclusions are supported by results. However, I am not sure if Vaccines is the best venue to publish these results. Before all, the topic is very interesting, and, undoubtedly, the paper is publishable but, Vaccines by MDPI is a medium-high impact factor journal (IF 2024: 3.4) and its broad spectrum of readers expects articles at edge of knowledge. I hope not to cause misunderstandings; the paper has enough merits and positive aspects to be published as I mentioned before (practically in its present form), but the scope is limited to its region of origin (Nigeria and near countries). Indeed, it may be expected that the paper will be low cited. Of course, the main findings are very interesting and important, but I leave the final decision to the Editor. For this reason, I would suggest major revision (not because the article needs it, at all). As the sole suggestion if the paper is accepted, authors could include catching-eye graphics into various parts of their manuscript.
Author Response
We are deeply grateful for your thoughtful evaluation and encouraging feedback on the quality and relevance of our manuscript. We particularly appreciate the acknowledgment that the study is well written, methodologically sound, and publishable in its present form.
While the data are indeed country-specific, the implications of unresolved vaccination inequity are far-reaching and transcend national borders. Persistent inequities in routine immunization coverage not only undermine progress toward global health security but also create fertile ground for outbreaks of vaccine-preventable diseases, which can spread rapidly across regions and continents, as seen in the 2023 Diphtheria outbreak in north-western Nigeria that spread into other regions of the country and even into neighbouring countries.
Moreover, inequitable vaccine access contributes to the inefficient utilization of limited and dwindling global health resources, as donor funding and emergency interventions are repeatedly redirected to address preventable crises in underserved regions. By providing rigorous localized evidence, this study highlights systemic vulnerabilities that are highly relevant to the global community of researchers, policymakers, and funders engaged in advancing immunization equity and resilience.
Finally, we emphasize that shying away from country- or region-specific studies risks perpetuating the systemic absence of critical literature on affected regions within the global body of scientific knowledge. Journals like Vaccines, with their international readership and commitment to equity, play a crucial role in ensuring that such evidence is integrated into the global discourse, thereby fostering inclusive and evidence-based solutions.
In light of this, we believe that our manuscript aligns with the journal’s mission and will be of interest to a wide readership concerned with immunization equity, program resilience, and global health security.
We have also included map-based illlustrations to the revised manuscript to improve visual clarity for the readers.
We hope these responses have addressed your concerns about our manuscript.
Thank you again.
Round 2
Reviewer 4 Report
Comments and Suggestions for Authors
The paper has been enhanced. Thus, I reccommend its publication practically in its present form.
Author Response
Thank you very much for the feedback. Very much appreciated!